# Assessing the Risk of Exposure to Aflatoxin B1 through the Consumption of Peanuts among Children Aged 6–59 Months in the Lusaka District, Zambia

**DOI:** 10.3390/toxins16010050

**Published:** 2024-01-17

**Authors:** Grace Musawa, Flavien Nsoni Bumbangi, Chisoni Mumba, Branly Kilola Mbunga, Gladys Phiri, Vistorina Benhard, Henson Kainga, Mkuzi Banda, Enock Ndaki, Ethel Mkandawire, John Bwalya Muma

**Affiliations:** 1Department of Disease Control, School of Veterinary Medicine, University of Zambia, Lusaka P.O. Box 32379, Zambia; musawagrace27@gmail.com (G.M.); cmumba@unza.zm (C.M.); vbenhard@hotmail.com (V.B.); enockndaki5@gmail.com (E.N.); ethel.mkandawire@unza.zm (E.M.); jmuma@unza.zm (J.B.M.); 2Lusaka District Health Office, Ministry of Health, Lusaka P.O. Box 50827, Zambia; 3School of Medicine, Eden University, Lusaka P.O. Box 37727, Zambia; 4Kinshasa School of Public Health, Faculty of Medicine, University of Kinshasa, Kinshasa 834, Congo; mbunga.branly@gmail.com; 5Partners in Food Solutions, Lusaka 10101, Zambia; gladysmphiri@gmail.com; 6Department of Veterinary Epidemiology and Public Health, Faculty of Veterinary Medicine, University of Agriculture and Natural Resources, Lilongwe 207203, Malawi; kaingahenson@yahoo.com; 7Zambia Compulsory Standards Agency, Lusaka P.O. Box 31302, Zambia; bandamkuzi@gmail.com

**Keywords:** aflatoxin B1, exposure risk assessment, peanut, under-five children, Zambia

## Abstract

Aflatoxins B1 (AFB1) are fungi-produced toxins found in crops like peanuts, maize, and tree nuts. They constitute a public health concern due to their genotoxic and carcinogenic effects. A deterministic exposure risk assessment to AFB1 through the consumption of peanuts was conducted on children using the Margin of Exposure (MOE) and the liver cancer risk approaches. Data on AFB1 concentrations in peanuts, quantities of peanut consumption, and the weights of the children were obtained from the literature. Generally, MOE values were below the safe margin of 10,000, ranging between 3.68 and 0.14, 754.34 and 27.33, and 11,428.57 and 419.05 for the high (0.0466 ng/kg), median (0.00023 ng/kg), and low (0.000015 ng/kg) AFB1 concentration levels, respectively. The liver cancer risk upon lifetime exposure to highly AFB1-contaminated peanuts (0.0466 ng/kg) ranged between 1 and 23 (95% lower bound) and 2 and 50 (95% upper bound) cases in a million individuals: a public health concern. A low liver cancer risk (≤1 case in a billion individuals upon lifetime exposure) was shown at median and low AFB1 concentrations. However, the risk of AFB1 should be a priority for risk management since its harmful effects could be potentiated by poor diet, high malnutrition levels, and other disease burdens in Zambia’s children.

## 1. Introduction

Aflatoxin B1 (AFB1) is a secondary toxic metabolite produced by *Aspergillus flavus* and is abundant in warm and humid regions of the world [1,2]. AFB1 can contaminate crops in the field, at harvest, and during storage [3]. AFB1 is a toxin of public health significance due to its toxic, genotoxic, and carcinogenic effects in animals and humans [4,5,6]. AFB1 poisonings have been reported, including the most severe one in 2004 in Kenya, where 125 people died [1]. In addition, chronic harmful effects of AFB1 have been documented, with a more significant impact on immunocompromised persons and children [7,8].

Human exposure to AFB1 results from both the level of contamination in a commodity and the quantity of the commodity consumed [9]. Commodities such as peanuts and maize are highly susceptible to AFB1 and highly consumed by the Zambian population [3]. In Zambia, a study by [10] revealed a high level of AFB1 in peanuts sold in supermarkets and open markets in the Lusaka District. However, to meet the children’s protein requirements, most parents feed their children peanut porridge since peanuts are considered a cheap source of protein [11]. Furthermore, most peanut porridge is mixed with maize meal, another highly susceptible crop to AFB1 contamination [3]. Furthermore, studies have reported poor dietary diversity in Zambian children’s nutrition habits, mostly centred around peanuts and maize meals [12]. This increases the consumption of these contaminated foods, exposing children to the harmful effects of AFB1.

In contrast to the considerable consumption of peanuts, there is a paucity of literature quantifying the potential risk of developing deleterious chronic effects of AFB1 associated with these commodities in Zambian children. These data are important because they help risk managers develop effective mitigation strategies and contingency plans. Dietary risk assessment is a widely endorsed approach to evaluating contaminants in food. It uses a conceptual framework where the hazard is identified and characterized, the exposure to the hazard evaluated, and the risk effects of the hazard in the targeted population are characterized [13]. Carrying out an exposure assessment in food safety entails estimating the possible dietary intake of a contaminant and characterizing its risk, using probability estimates of health outcomes to occur in a population under defined exposure conditions [14].

Considering that children are the most susceptible to AFB1 due to their greater exposure to contaminated food, their immature metabolism and elimination system, and their higher growth and development rates [15], it was necessary to estimate the risk of exposure to AFB1 among children of the Lusaka District, as a result of the consumption of peanuts, and assess the risk of liver cancer associated with the aforementioned exposure.

## 2. Results and Discussion

### 2.1. Contamination and Consumption

Raw peanut samples from open markets and supermarkets were collected in Lusaka District and analysed for AFB1 presence and concentration levels (Table 1). Overall, 55.4 percent of all samples tested positive for AF presence, with concentration levels ranging from 0.014 to 48.67, a median of 0.23 ppb, and a geometric mean ± SD of 0.43 ± 9.77 ppb. AFB1 had the highest concentration levels (46.6 ppb), median (0.23 ppb), and mean concentration (0.45 ± 9.41 ppb).

AFB1 intake was estimated using the average consumption of raw peanuts used to prepare porridge for children (Table 2) [11], while the weights for the children were obtained from the WHO standard guidelines for growth monitoring for children under five (Table 3).

### 2.2. Risk Characterization

#### 2.2.1. The Margin of Exposure (MOE) Approach

Three scenarios were built during the estimation of the MOE using the high, median, and low concentrations of AFB1 in peanuts. In all the scenarios, the FFC ranged from once a week to twenty-one times per week. All MOE values in the three scenarios were below the safe margin of 10,000, indicating a risk management priority, except for the age category between 24 and 59 months in the low AFB1 concentration levels and a FFC of once per week, where the MOE was 11,428.57. The MOE ranged between 3.68 and 0.14, 754.34 and 27.33, and 11,428.57 and 419.05 for the high (0.0466 ng/kg), median (0.00023 ng/kg), and low (0.000015 ng/kg) concentration levels of AFB1, respectively (Figure 1). The high AFB1 content in peanuts intended for children’s peanut porridge and their high consumption frequency are a great public health concern because they drastically reduced the MOE value, hence increasing the risk of cancer in the population.

#### 2.2.2. The Quantitative Liver Cancer Risk Approach (HCC)

The consumption of highly (0.0466 ng/kg) AFB1-contaminated peanuts revealed a high risk of developing liver cancer among children using the quantitative liver cancer risk approach estimate. This risk increased as the consumption frequency of AFB1-contaminated peanuts increased (Figure 2). The number of liver cancer cases could range from 0.09 to 2.3 cases per 100,000 individuals in 65 years (95% LB) and from 0.2 to 5.0 cases per 100,000 individuals in 65 years (95% UB). This implies that 1 to 23 (95% LB) and 2 to 50 (95% UB) individuals in a million upon lifetime exposure to AFB1 in peanuts would develop liver cancer. Although the estimated number of liver cancer cases at 95% LB seems lower for the low consumption frequency (once a week) of highly AFB1-contaminated peanuts, the liver cancer risk could be magnified by the effect of other highly consumed cereals like maize.

The risk of developing liver cancer due to the consumption of median (0.00023 ng/kg) and low (0.000015 ng/kg) amounts of AFB1-contaminated peanuts in children’s porridge was very low. The highest estimate of 0.0001 cases of liver cancer/100,000 individuals in 65 years is depicted in Figure 3. This implies one individual in a billion upon lifetime exposure, and is thus of low public health concern.

Regardless of the scenario used for these estimates, it was noted that the risk of cancer increased with the high consumption frequency of AFB1-contaminated peanuts. Even where the risk of liver cancer is low, the lack of diversity in the Zambian diet mostly centred around maize-based food [11,17], a highly susceptible crop to AFB1 could exacerbate this risk. Additionally, grains, nuts and seeds, and roots and tubers are present in 70 percent, 60 percent, and more than 80 percent of Zambian children’s diets, respectively [18]. Considering that peanuts are not eaten in isolation, the synergy of these susceptible foods magnifies the human health risk due to AFB1 in the Zambia population.

The cancer risk estimates reported in this study are of great public health concern since these estimates could have been overshadowed by the low prevalence of HBV (1.5 percent) used in the simulations. The prevalence of HBV reported by J Goma [19] was obtained from a restricted and small sample of children frequenting a tertiary hospital and one peripheral health centre. A prevalence of HBV drawn from a large and representative sample of Zambian children could depict the true scenario of the hepatocarcinoma risk in this age group.

Furthermore, AFB1’s chronic effects are exacerbated by conditions such as immunosuppression and malnutrition [20]. The high prevalence of HIV/AIDS in Zambia, including in children [21], coupled with the burden of malnutrition in children [12], predisposes these children to an increased cancer risk resulting from AFB1 exposure through peanut consumption. Additionally, being classified as a low-income country, Zambia is still battling poverty eradication. The high poverty level is a barrier to accessing quality and diversified foods. Most families depend upon a monotonous and unbalanced diet, worsening their nutritional status, which in turn exposes them to various diseases, resulting in a reduced capacity to generate more income, increasing the poverty level, and thus creating a vicious cycle favourable to the harmful effects of AFB1. The exposure risk could be much higher in children since peanuts are being promoted in undernourished children as a cheap source of protein to improve their nutritional status [22].

Strict regulation on the AFB1 levels in peanuts sold to the public is, therefore, necessary to avoid further exposure to children because of their vulnerability. This can only be effective using a multi-intervention approach considering the complexity of the problem. A farm-to-folk approach would be necessary to effectively reduce exposure to AFB1. At the production level, zero or low AFB1 contamination of peanuts and other crops should be ensured. At the value-addition, storage, and sale levels, the development and accumulation of AFB1 should be prevented. Regulatory agencies should ensure rapid testing of these food products destined for consumption. Finally, at a national level, policies that favour compliance to the set standards should be enforced along with creating an environment that reduces the burden of diseases and poverty levels, amplifying the chronic harmful effects of AFB1.

## 3. Uncertainty

There was little literature on AFB1 levels in peanuts in Zambia. Furthermore, peanuts are not the only food source of AFB1 exposure in children. The uncertainty of the prevalence of HBV could also affect the risk estimate. Furthermore, grey literature was used to estimate the risk of exposure to AFB1. Considering the above, the uncertainty associated with the estimate could be medium. Consequently, the risk estimates in the study should be considered with caution.

## 4. Conclusions and Perspectives

This study has revealed an increased exposure risk to AFB1, thereby increasing the risk of developing liver cancer in children under five who frequently consume AFB1-contaminated peanuts. The situation could be alarming if the simulation included other factors such as the mixed diet with maize meal, the high level of malnutrition, and several HBV prevalence estimates. Although the data on the concentration of AFB1 in peanuts came from one paper [10], this study is the first carried out in Zambia and could serve as a baseline for other robust risk assessment studies that could include maize, rice, cassava, and their products. It would also be important to extend the risk assessment of AFB1 in the above-susceptible crops to other parts of Zambia to ascertain the magnitude of the problem at a national level that could trigger a priority for risk management.

The high liver cancer risk reported from this study is a wake-up call for all stakeholders involved in mitigating the harmful effects of aflatoxins associated with these susceptible crops. We, therefore, recommend strict adherence to good agricultural practices (GAPs), good manufacturing practices (GMPs), and good hygiene practices (GHPs), which are fundamental in reducing the formation, accumulation, and contamination of aflatoxins in susceptible crops in the pre- and post-harvest stages. This will protect the public and prevent economic losses. Furthermore, government agencies should strengthen border control by ensuring strict compliance to the maximum permissible limits (MPLs) set by the Codex Alimentarius Commission for all crops susceptible to aflatoxins being imported into the country. Finally, public awareness of the potential health risks of aflatoxins is critical in mitigating their occurrence in susceptible crops.

## 5. Materials and Methods

### 5.1. Peanuts Sampling

Secondary data on AFB1 concentration in peanuts sold in both open and supermarkets in Lusaka Districts [10] were used in this study. In their cross-sectional study, [10] sampled 32 markets, including 26 open markets and 6 supermarkets, across the seven constituencies of Lusaka District. A simple random sampling was used from each open market to select at least 3 vendors, or 10% of them if the number was large. From each selected vendor of the open market and each supermarket, 500 g of raw peanuts of each variety were purchased. In total, 92 raw peanut samples were purchased from open markets (n = 73) and supermarkets (n = 19) [10].

### 5.2. Aflatoxins Extraction and Determination

Peanut samples were analysed in the chemistry laboratory at Zambian Agriculture Research Institute (ZARI) using an AflaTest^®^ test kit with high-performance liquid chromatography (HPLC) method certified by the AOAC^®^ Official Methods Programme, as the official method 991.31 applicable for the determination of aflatoxin B1, B2, G1, and G2 both by fluorometry and HPLC analysis in corn, peanuts, and peanut butter. HPLC-grade reagents were used to run the high-performance liquid chromatography, while acetonitrile and methanol were purchased from Sigma-Aldrich^®^ (Darmstadt, Germany). Aflatoxin B1, B2, G1, and G2 standards were purchased from Trilogy Analytical Laboratory (Washington, DC, USA) (Lot 120316-090, Total concentration AF: 5.0 mg/mL, Total aflatoxin B1, B2, G1, G2: 4/1/4/1). The concentration was determined according to AOAC International Official Methods of Analysis. The AflaTest^®^ column (Vicam, Watertown, MA, USA) was used as an immunoaffinity column (IAC) to clean the samples.

Each peanut sample (500 g) was entirely ground to minimise the sub-sampling error in AF analysis using a domestic grinder (Jura-CAPRESSO INC, Model N° 503, Shanghai, China). Then, 25 g of each ground sample was mixed with 5 g NaCl and placed in a blender jar for extraction using 125 mL of methanol/water (70:30). The solution was blended at high speed for 2 min and then filtered using fluted filter paper (Whatman No. 4). The extract was diluted with 30 mL of purified water before being filtered through a glass microfiber filter into a clean vessel.

AflaTest^®^ immune-affinity columns (IACs) were used to clean up the samples. Fifteen millilitres of the filtrate diluted extract was passed through the AflaTest^®^ IAC at a rate of 1–2 drops/second until air came through the column. The column was then washed twice with 10 mL of purified water at a rate of 2 drops/second, and the glass cuvette (VICAM part # 34000) was placed under AflaTest^®^ IAC and 1.0 mL of HPLC grade methanol was added into a glass syringe barrel. Finally, AflaTest^®^ IAC was eluted at a rate of 1 drop/s by passing the methanol through the column, and all the sample eluate (1.0 mL) was collected in a glass cuvette. An additional 1.0 mL of purified water was poured to eluate and analysed by HPLC.

Reverse-phase HPLC was used to quantify AFs along with a fluorescence detector followed by post-column derivatization (PCD) involving bromination using a water HPLC system (pump 1525; fluorescence detector 2475; analytical column Nova-pack-C18250 × 4.6 mm: 5 mm). Kobra cell was used, and bromide was added to the mobile phase to achieve PCD. Fifty microliters of diluted AF eluate were then injected into HPLC. The mobile phase included water, methanol, and acetonitrile mixture with a 600:300:200 (*V*/*V*/*V*) ratio. A sample was considered positive for AF if at least one of the four types was positively observed on the HPLC chromatogram reading.

The limit of detection using the protocol described above was 0.10 ppb for total aflatoxin and 0.05 ppb for AFB1, 0.03 ppb for AFB2, 0.03 ppb for AFG1, and 0.05 ppb for AFG2. The correlation coefficient (r) of 0.9951 from the linear regression equation (Y = 0.9242X + 0.0547) indicated that the linearity of this method was excellent. The percentage recovery from this method was greater than 85% for aflatoxin B1, B2, and G1 from 0.10 ppb to 10.0 ppb. Aflatoxin G2 recovery was greater than 55% for the 0.5–10 ppb range. The validation data from the method used showed very good reproducibility for total aflatoxin (ranging from 0.5 ppb to 10 ppb) with an average coefficient of variation of less than 8%. For individual aflatoxins B1, B2, G1, and G2 (ranging from 0.5 ppb to 10 ppb), the average coefficient of variation was less than 12%. The method also had excellent repeatability with a coefficient of variation for total aflatoxin of 2.6% and less than 3.2% for individual aflatoxin B1 and B2.

### 5.3. Determination of the Exposure Risk to AFB1

To determine the exposure risk to AFB1, we used the quantitative liver cancer risk approach [23] and the Margin of Exposure (MOE) approach for compounds that are both genotoxic and carcinogenic [24]. Furthermore, all the calculations assumed that the following: (1) the amount of AFB1 in the peanuts was equal to the amount contained in the porridge; (2) heat and storage had no effect on the concentration of AFB1 in peanuts. Except for differences in exposure to hepatitis B virus (PHBsAg^+^ or PHBsAg^−^), we further assumed no differences in the health status, the enzymatic activity of the liver, and the dimensions and weight through sound waves in the hypothetical study population.

#### 5.3.1. Hazard Identification and Characterization

Aflatoxins are a group of cancer-causing mycotoxins and toxic metabolites produced by *Aspergillus flavis*, *Aspergillus parasiticus*, and *Aspergillus nomius* [25,26]. They mostly contaminate cereals, grains, peanuts, seeds, and legumes [27]. Among the identified types of aflatoxins, AFB1 is the most toxic [28], with mutagenic, carcinogenic, and potential teratogenic effects [29]. This has led the International Agency for Research on Cancer (IARC) to classify AFB1 as carcinogenic in humans [30]. It has been established that AFB1 exposure increases the risk of liver cancer in humans by inducing DNA adducts that lead to a genetic change in liver cells [31]. This risk is further increased in individuals with chronic hepatitis B virus (HBV) infections [20,32].

#### 5.3.2. Exposure Assessment

Codex Alimentarius Commission (CAC) recommends an intake of aflatoxin B, G, and M that is as low as reasonably possible [23]. In Zambia, the maximum allowable limit for AFB1 is 15 ppb [3], although this is not based on risk assessment data. The consumption frequency, the quantity of food commodities, and the concentration of aflatoxin data are important components of the exposure risk assessment (Figure 4).

The dietary exposure evaluation was calculated using Equation (1) of the Estimated Daily Intake (EDI) [23], as shown below:EDI = (C × CR)/BW(1)
where C is the concentration (ng/kg) of the mean AFB1 concentration in the raw peanut [10]; CR is the food item’s daily consumption rate (g/kg BW/day); and BW is the average body weight of the age group. The daily consumption rate of raw peanuts was obtained from the Zambia Food-Based Dietary Guidelines (FBDG) Technical Recommendation in 2021 [11].

#### 5.3.3. Risk Characterization

The risk characterization was conducted using two methods of the deterministic approach established by international regulatory agencies: the Margin of Exposure (MOE) recommended by the EFSA [24] and the quantitative liver cancer risk approach recommended by the FAO (Québec City, QC, Canada) and the WHO (Geneva, Switzerland) [23].

##### The Margin of Exposure (MOE)

The margin of exposure (MOE) approach gives the ratio between the reference dose level that causes a 10% increase in the incidence of cancer in rodents (BMDL_10_) and the estimated daily intake (Equation (2)) [24].
MOE = BMDL_10_/EDI(2)

A BMDL_10_ of 0.4 ng/kg bw/day estimated by EFSA based on carcinogenicity data in rats exposed to AFB1 was used in this study. MOE of 10,000 or greater has no public health concern. However, a value lower than 10,000 indicates a high-risk management priority due to potentially harmful human health effects [24].

##### Quantitative Liver Cancer Risk Approach

The risk of liver cancer resulting from AFB1 exposure was calculated using Equation (3) and expressed in the number of cases/year/100,000 individuals.
(3)$$\mathrm{Cancer}\;\mathrm{risk}=P_{cancer}\times \mathrm{EDI}$$

The carcinogenic potency ($P_{cancer}$) of AFB1, expressed in ng/kg bw/day, was calculated using Equation (4). This estimation considers the carcinogenic potency for individuals with hepatitis B virus (PHBsAg^+^), which has been established at 0.017 cases/year/100,000 individuals per ng of exposure to AFB1; the $P_{cancer}$ for non-infected individuals (PHBsAg^−^) established at 0.269 cases/year/100 000 individuals per ng of exposure to AFB1; the prevalence of carriers of hepatitis B virus (%PHBsAg^+^); and non-HBV carriers (%PHBsAg^−^). The 95% lower bound (LB: 0.017–0.269) and upper bound (UB: 0.049–0.562) estimates for HBV antigen-negative and antigen-positive individuals determined by JECFA [23] were considered. This study estimated the prevalence of HBV carriers in children under five at 1.5% [19]. We assumed that there were no differences in the health status, the enzymatic activity of the liver, and the dimensions and weight through sound waves of our study population.
(4)$$P_{cancer}=({\mathrm{PHBsAg}}^{+}\times \%\mathrm{population}\;{\mathrm{PHBsAg}}^{+})+({\mathrm{PHBsAg}}^{-}\times \%\mathrm{population}\;{\mathrm{PHBsAg}}^{-})$$

The $P_{cancer}$ equation only considers one year’s estimated number of liver cancer cases. Therefore, to obtain the risk of lifetime exposure, we converted this risk into a risk originating from the Zambian lifetime exposure (65 years) by multiplying the calculated $P_{cancer}$ by 65 years [33]. A safe margin of 0.1 cancer cases/100,000 individuals per 65 years was considered an acceptable number and of non-concern for risks associated with food consumption. This criterion implies one additional cancer case in one million individuals in lifetime exposure.

### 5.4. Statistical Analysis

Data on the concentration of aflatoxins in peanuts, the food frequency consumption, and the weight of children under five were summarized in Excel for Microsoft Windows (version 16). The average weight of each under-five-year category was calculated from the standard weight of WHO. In contrast, the median, minimum, and maximum concentrations of aflatoxins were calculated using Stata (StataCorp, College Station, TX, USA) version 16.0 for Windows. The Estimated Dietary Intake, MOE values, carcinogenic potency ($P_{cancer}$) of AFB1, and cancer risk were calculated, and graphs were generated from Excel 2016^®^. 

## Figures and Tables

**Figure 1 toxins-16-00050-f001:**
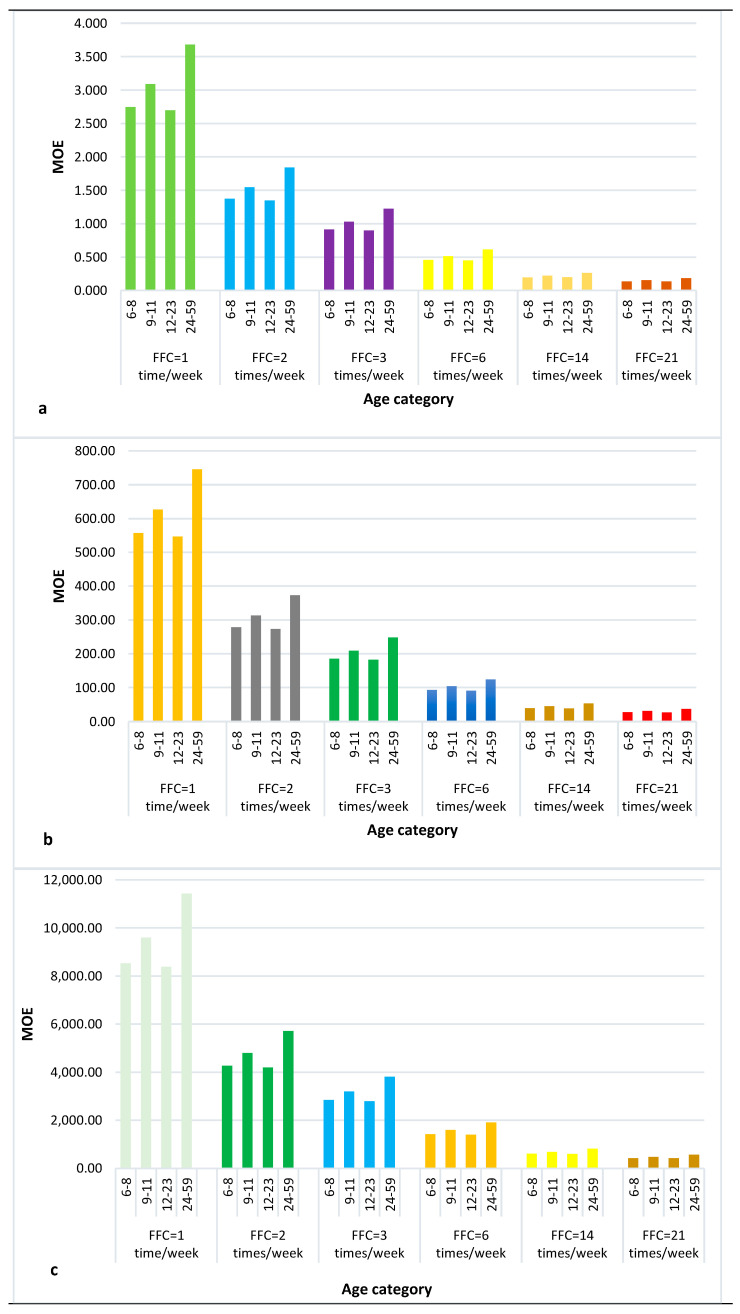
Estimation of MOE in relation to the FFC with (**a**) high, (**b**) median, (**c**) low AFB1 concentration in peanuts.

**Figure 2 toxins-16-00050-f002:**
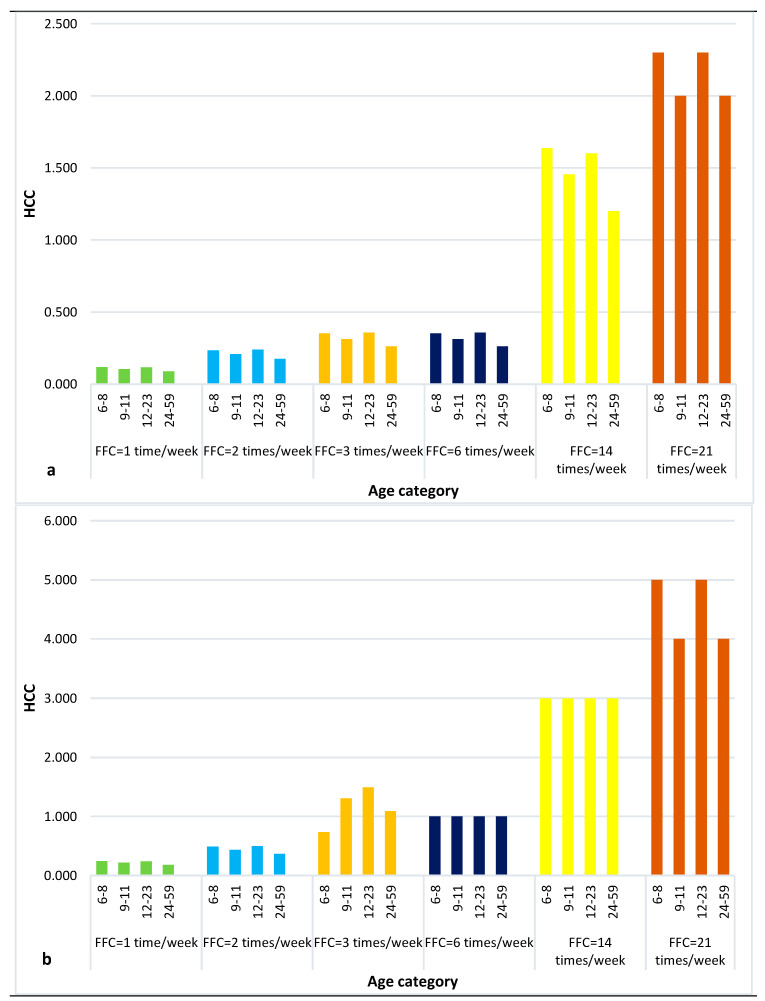
Estimation of HCC in relation to the FFC with high AFB1 concentration in peanut established at 95% (**a**) LB and (**b**) UB.

**Figure 3 toxins-16-00050-f003:**
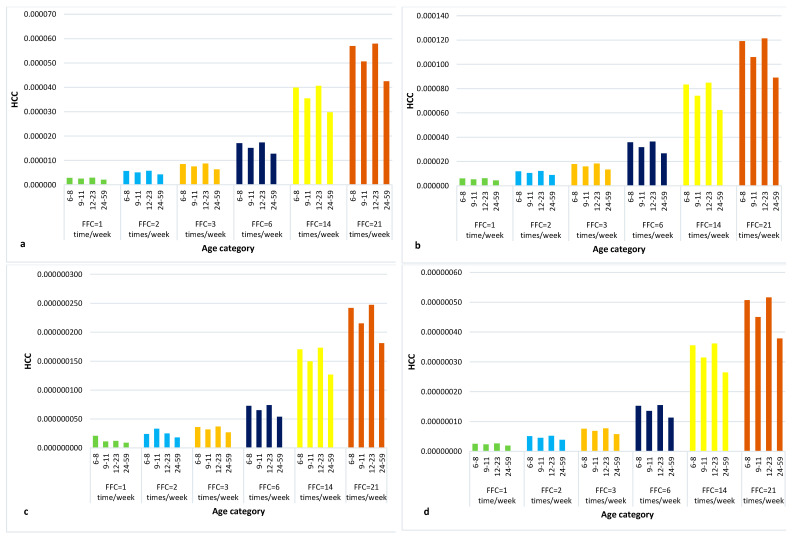
Estimation of HCC in relation to the FFC with median AFB1 concentration in peanuts established at 95% (**a**) LB and (**b**) UB, and low AFB1 concentration in peanuts established at 95% (**c**) LB and (**d**) UB.

**Figure 4 toxins-16-00050-f004:**
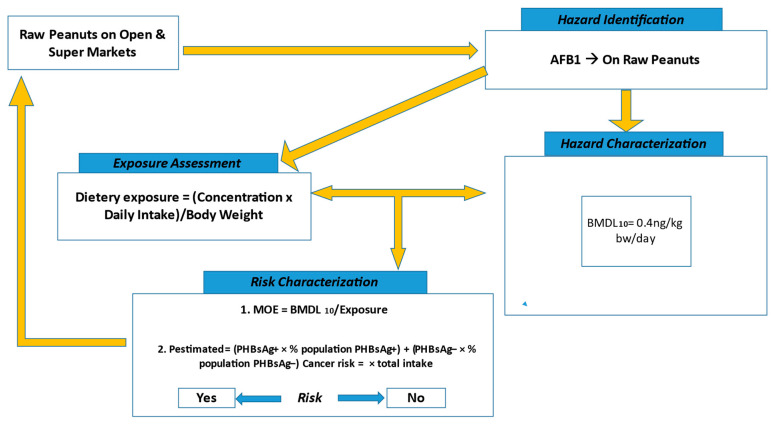
Flowchart of dietary exposure estimates for the AFB1 in peanuts sold in both open and supermarkets. Note: The dose/effect curve was not established in this study.

**Table 1 toxins-16-00050-t001:** Summary of aflatoxin concentration in raw peanut samples from the Lusaka District.

Variables	Positive (%)	Median (ng/kg)	Min (ng/kg)	Max (ng/kg)
AFB1	n = 41(44.6)	0.00023	0.000015	0.0466
AFB2	n = 41(44.6)	0.000132	0.000006	0.01317
AFG1	n = 21(22.8)	0.000028	0.000005	0.00051
AFG2	n = 7(7.6)	0.000008	0.000006	0.00004
AF	n = 51(55.)	0.00023	0.000014	0.04867

Source: [10].

**Table 2 toxins-16-00050-t002:** Summary of the food frequency consumption (FFC) by age in Zambian children (2021).

Child’s Age in Months	Frequency of Consumption per Day	Amount of Food ** (Porridge)
6–8	2–3	2–3 tablespoons *
9–11	3–4	2–3 tablespoons
12–23	3–4	3–4 tablespoons
24–59	4–5	3–4 tablespoons

Source: [11] * One tablespoon = 10 g; ** Amount of food per consumption.

**Table 3 toxins-16-00050-t003:** Standard and average weight of children under five by age.

Age (Months)	Weight (Kg)		
	Female	Male	Average
6–8	7.3–8.2	7.9–8.9	8.1
9–11	8.7–9.4	9.5–10.2	9.4
12–23	9.0–11.3	9.7–12.0	11.0
24–59	12.1–18.0	12.7–18.5	15.3

Source: [16].

## Data Availability

The essential data supporting the reported results are contained in this study.

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
