# Peer review of "Assessing the Risk of Exposure to Aflatoxin B1 through the Consumption of Peanuts among Children Aged 6–59 Months in the Lusaka District, Zambia"

_toxins, 2024, doi:10.3390/toxins16010050_

Round 1

Reviewer 1 Report

Comments and Suggestions for Authors

Kindly state sumary of results in the abstract, and decrese the rivew in it 

State the extraction method of AFs in the methodology, and all methods you used in your study  

State the results to be more clearly 

Arrange all segments of the paper to be clear to reader 

After that send me the paper in word formate 

Comments on the Quality of English Language

Kindly state sumary of results in the abstract, and decrese the rivew in it 

State the extraction method of AFs in the methodology, and all methods you used in your study  

State the results to be more clearly 

Arrange all segments of the paper to be clear to reader 

After that send me the paper in word formate 

Reviewer 2 Report

Comments and Suggestions for Authors

Dear authors,

I have enclosed herewith my review report.

Comments on the Quality of English Language

It needs a moderate editing by a native English speaker.

Reviewer 3 Report

Comments and Suggestions for Authors

Dear authors

Age category needs more clarification in all the histograms and tables presented in the manuscript. The first three were in order: 1/day, 2/day, 3/day, while the next three were 6/week, 14/week, 21/week, i.e. almost the same if they were divided by the number of days of the week. . This is despite the contradictory results presented for each group.

Although aflatoxins target the liver, the study did not pay attention to the difference in health status and enzymatic activity of the liver, as well as the dimensions and weight through sound waves of those under study and before starting it.

The methods, tools, and mechanism for collecting data are not completely clear.

The planning of the experiments and the statistical analysis of the evidence and results are unclear, which makes these results mere observations that cannot be relied upon.

There is no clear recommendation or conclusion indicated by this study, despite the importance of the topic at hand, which requires reformulating it in another way.

Comments on the Quality of English Language

Dear

I think the language is very clear and understandable, although it may need a quick review from a fluent English speaker before publishing. 
